# Effect of Moderately High-Dose Vitamin D3 Supplementation on Mortality in Patients Hospitalized for COVID-19 Infection

**DOI:** 10.3390/nu17030507

**Published:** 2025-01-30

**Authors:** Liza Dalma Sümegi, Marina Varga, Veronika Kadocsa, Balázs Szili, Márk Stempler, Péter András Lakatos, Zsuzsanna Németh, István Takács

**Affiliations:** 1Department of Internal Medicine and Oncology, Semmelweis University, Korányi S. U 2/a, 1083 Budapest, Hungary; sumegi.liza.dalma@semmelweis.hu (L.D.S.); szili.balazs@semmelweis.hu (B.S.); stempler.mark@semmelweis.hu (M.S.); lakatos.peter@semmelweis.hu (P.A.L.); nemeth.zsuzsanna@semmelweis.hu (Z.N.); 2Department of Laboratory Medicine, Semmelweis University, Nagyvárad Tér 4, 1089 Budapest, Hungary; varga.marina@semmelweis.hu; 3Department of Pulmonology, Semmelweis University, Tömő U. 25–29, 1083 Budapest, Hungary; kadocsa.veronika@stud.semmelweis.hu

**Keywords:** vitamin D, moderately high-dose supplementation, COVID-19, mortality

## Abstract

Background: Despite a large number of published studies, the effect of vitamin D3 supplementation on mortality in hospitalized patients, as well as the recommended dose and duration of therapy, is unclear. In our retrospective study, we aimed to investigate the impact of vitamin D deficiency and moderately high-dose vitamin D3 supplementation on mortality and disease outcomes in patients with COVID-19 infection. Methods: We analyzed data from 148 COVID-19-infected hospitalized patients in two different departments, Internal Medicine and Oncology, at Semmelweis University. The severity of COVID-19 and the treatment used were the same except at one of the departments, where patients received circa 90,000 IU of vitamin D3. We compared in-hospital mortality rates between the groups. In a subgroup analysis, we evaluated the efficacy and safety of vitamin D3 supplementation by assessing 25(OH)D and 1,25(OH)_2_D concentrations on days 0, 4, and 8. Results: As a result of the supplementation, the deficiency was resolved in 4 days in deficient patients, and none of the 25(OH)D or 1,25(OH)_2_D concentrations exceeded the normal range. Mortality was significantly lower and decreased 67% in the group receiving vitamin D3 supplementation, regardless of baseline 25(OH)D concentrations. Conclusions: The supplemental dosage, 3 × 30,000 IU of vitamin D3, is effective and safe and may reduce mortality in COVID-19 infection.

## 1. Introduction

There is an increasing amount of evidence regarding the non-osteometabolic effect of vitamin D, particularly its role in modulating immune responses [1,2,3]. The discovery of the vitamin D receptor (VDR) in various types of immune cells, such as monocytes, dendritic cells, and activated T cells, suggested a role for vitamin D in modulating immune functions [4,5]. Vitamin D plays a role in the defense against many viral infections (e.g., HIV1, Rota, and Hepatitis C) as well [6,7,8]. Vitamin D3 supply contributes to the balanced functioning of the immune system, for example, in the case of a COVID-19 infection [9]. However, according to other studies, systemic inflammation associated with COVID-19 infection can cause lower 25(OH)D concentrations [10,11]. Nonetheless, vitamin D can play a role in warding off the cytokine storm [9] because vitamin D inhibits Th1 cell responses that generate proinflammatory cytokines like INFγ and TNFβ [12].

Regular bolus cholecalciferol supplementation has been associated with less severe COVID-19 and improved survival among patients treated in geriatrics [13]. A study by Entrenas Castillo M et al., on hospitalized patients verified that calcifediol supplementation—in addition to the best available therapy—reduced disease severity and intensive care unit (ICU) treatment among SARS-CoV-2 positive patients [14]. Other Spanish studies show that calcifediol, which is available in Spain, increases 25(OH)D concentrations more rapidly than cholecalciferol and improves disease outcomes [15,16,17]. In a randomized, placebo-controlled trial conducted by Rastogi A et al., asymptomatic or mildly symptomatic patients who tested positive for SARS-CoV-2 and had an initial 25-hydroxyvitamin D (25(OH)D) concentration of less than 20 ng/mL were assigned to two groups. One group received a daily supplement of 60,000 IU of vitamin D3 for 7 days, aiming to elevate their 25(OH)D concentration to 50 ng/mL. The findings indicated that vitamin D-deficient patients with COVID-19 experienced a higher rate of conversion to SARS-CoV-2-negative status compared to the control group following this high-dose vitamin D3 supplementation [18]. A study by Sánchez-Zuno GA et al., found that among 42 outpatients, those who received 10,000 IU of vitamin D3 for 14 days reported fewer COVID-19 symptoms, with normalized 25(OH)D concentrations by day 14 [19].

Despite the encouraging results of published studies, meta-analyses have found conflicting results regarding the effectiveness of vitamin D supplementation on the severity and mortality of COVID-19 infection [20,21,22,23,24]. A meta-analysis by Tentolouis et al., found that vitamin D3 supplementation may reduce ICU admission but has no effect on hospital mortality [25]. According to a meta-analysis by Pal et al., vitamin D3 supplementation was associated with reduced mortality; however, the optimal dose and duration of administration remain uncertain [26].

Due to preceding controversial findings and the unclear dosing requirements in our retrospective study, we aimed to investigate the effect of short-term, moderately high-dose vitamin D3 supplementation, approximately 90,000 IU, on mortality in patients hospitalized for COVID-19 infection. Additionally, by measuring the concentration of 25(OH)D and active vitamin D (1,25(OH)_2_D), we aimed to assess the effectiveness and safety of supplementation in both vitamin D-deficient (defined as serum 25-hydroxyvitamin D (25(OH)D) < 50 nmol/L or < 20 ng/mL) and -non-deficient patients within a hospital setting.

## 2. Materials and Methods

### 2.1. Study Design

We conducted a two-center retrospective study between December 2022 and April 2023 in which patients diagnosed with COVID-19 infection were randomly selected by the staff upon admission to the emergency department after assessing disease severity for each patient using the National Early Warning Score 2 (NEWScore 2) [27]. Patients with severe disease, who needed intensive respiratory support, were transferred to the ICU, while the remaining patients were then assigned to either Department 1 (Dept1) or Department 2 (Dept2) of Internal Medicine and Oncology at Semmelweis University. In Dept1, mandatory vitamin D3 supplementation was introduced early as part of the local protocol, and our patients received vitamin D3 from the first day, whereas in Dept2, all patients received vitamin D3 supplementation at a later stage, after we completed our study; they did not receive vitamin D3 supplementation during our study. The other treatment protocols between the two departments were otherwise identical, during the study between December 2022 and April 2023, allowing for a retrospective analysis of the data.

The 25(OH)D concentrations were measured in all patients at admission (day 0), and mortality was documented in both departments. Chronic diseases were recorded according to established guidelines (List A1).

The primary objective of this study was to compare mortality rates among randomly selected patients between the two departments, with the only difference in patient care protocols being the administration of vitamin D3 supplementation. Additionally, we aimed to investigate the relationship between 25(OH)D concentration and other clinical variables to assess the impact of vitamin D3 supplementation on 25(OH)D and 1,25(OH)_2_D concentrations in a randomly selected subgroup of 30 patients supplemented in Dept1. In this subgroup, both 25(OH)D and 1,25(OH)_2_D concentrations were measured on days 0, 4, and 8.

In Dept1, all hospitalized COVID-19 patients received circa 90,000 IU of vitamin D3 supplementation (12,000 IU for 7 days and later with protocol changes; 30,000 IU daily for 3 days) based on the existing protocol, regardless of the initial 25(OH)D concentration on day 0. After the moderately high-dose supplementation, patients in Dept1 received a maintenance dose of 3000 IU of vitamin D3 during their hospital stay.

To monitor the safety of the supplementation, 25(OH)D, serum calcium, and phosphate concentrations were measured both at baseline and after supplementation. However, 24-h urine calcium measurement was not feasible due to infectious disease protocols in place during the pandemic.

This research was approved by the Regional and Institutional Committee of Science and Research Ethics of Semmelweis University (RKEB 245-1/2020), and all study-related procedures were conducted in accordance with the World Medical Association’s Declaration of Helsinki.

### 2.2. Participants and Study Criteria for Vitamin D3 Supplementation

A total of 148 patients were randomly selected: 76 patients in Dept1 and 72 patients in Dept2. A randomly chosen subpopulation of Dept1 (30 patients) was further investigated to assess the efficacy and safety of vitamin D3 supplementation.

The vitamin D3 tablets were commercially available preparations: 30,000 IU of vitamin D3 by Pharma Patent and 3000 IU of vitamin D3 by Bioextra. The supplementation was started at the time of admission if there were no contraindications (elevated serum calcium and phosphate concentration and intake of more than 1000 IU of vitamin D3 per day). Following the moderately high-dose supplementation, patients in Dept1 received a maintenance dose of 3000 IU of vitamin D3 for the duration of their hospital stay. The amount of vitamin D that was given was considered effective and safe, in accordance with international recommendations and our previous findings [28].

### 2.3. Laboratory Measurements

COVID-19 infection was confirmed upon admission using a rapid antigen test (Genedia W COVID-19 Ag, Cat No. 76-643K-21-1, Green Cross Medical Science, Geumwang-eup, Republic of Korea). If the clinical features of COVID-19 were unclear, a polymerase chain reaction (PCR) test was conducted to confirm the diagnosis.

The 25(OH)D concentrations of all patients were measured on day 0 immediately after blood collection, reported in ng/mL, using the LIAISON 25 OH Vitamin D TOTAL Assay (DiaSorin Inc., Stillwater, NM, USA) by the central clinical laboratory of the university hospitals (Department of Laboratory Medicine, Semmelweis University, Hungary). At this time, other relevant laboratory variables, such as calcium, phosphate, magnesium, creatinine, and kidney function, were also assessed at this time in the same laboratory using a rutin analyzer (Siemens Atellica CH 950, Tarrytown, NY, USA).

Additionally, blood samples, collected according to the COVID-19 protocol on days 0, 4, and 8 from a randomly selected subgroup of 30 patients, were processed (centrifuged for 15 min at 4 °C) to obtain serum samples for further analysis. These serum samples were then stored at −80 °C for a maximum of 30 days for the later determination of 25(OH)D and 1,25(OH)_2_D concentrations using a competitive chemiluminescent, fully automated immunoassay (25(OH)D—Bio-Rad EQAS External Quality Assurance Service QC system with CV: 6.10, Z-score: 0.78, %: 4.74; 1,25(OH)D—the data were extracted from the QC program of the Liason XL system, and the result was as follows: CV: 4.73%; CLIA, LIAISON analyzer, DiaSorin Inc., Stillwater, NM, USA).

### 2.4. Statistical Analysis

An unpaired *t*-test with Welch’s correction was used to compare 25(OH)D concentrations, while the Mann–Whitney test was used to compare NEWScore 2, the presence of chronic kidney disease (CKD), and the mortality of patients supplemented in Dept1 and Dept2. The chi-squared test was used to assess mortality in deficient and non-deficient patients in Dept1 and Dept2. Repeated-measures ANOVA with Dunnett’s test for multiple comparisons was applied to assess 25(OH)D and 1,25(OH)_2_D concentrations on days 0, 4, and 8 in the vitamin D-deficient (<20 ng/mL) and -non-deficient (≥20 ng/mL) groups of the 30 randomly selected patients supplemented in Dept1. These analyses were conducted using GraphPad Prism v8.0.2 software (GraphPad Software LLC, San Diego, CA, USA). Results were significant if *p* < 0.05.

## 3. Results

### 3.1. Patients’ Characteristics

The characteristics of the 148 patients included in this study are summarized in Table 1. Across the departments, 82 patients were male, and 66 were female. The study included 76 patients from Dept1 and 72 patients from Dept2. The mean age was 69 ± 16 years in Dept1 and 66 ± 14 years in Dept2. Among the patients admitted to the two departments, vitamin D deficiency was confirmed in a total of 54 cases (25(OH)D < 20 ng/mL), and in 94 patients, the 25(OH)D concentration was in the non-deficient range (25(OH)D ≥ 20 ng/mL). The number of patients with vitamin D deficiency (<20 ng/mL) did not differ significantly in the groups receiving and not receiving vitamin D3 supplementation (Dept1: 24, Dept2: 30 patients). There was no significant difference between the mean 25(OH)D concentrations measured at admission in the groups receiving or not receiving vitamin D3 supplementation (Dept1: 29 ng/mL; Dept2: 29 ng/mL) (Table 1, Figure 1a). No differences were found in the chronic diseases reported in the medical history or diagnosed at admission, including hypertension, type 1 diabetes (T1DM), type 2 diabetes (T2DM), non-ST-elevation myocardial infarction (NSTEMI), ST-elevation myocardial infarction (STEMI), stroke, chronic obstructive pulmonary disease (COPD), chronic heart disease (CHD), and various malignancies (Appendix A Table A1). The only exception that was noted was the higher incidence of chronic kidney disease (CKD) among patients in Dept1 (Dept1 median: 1, Dept2: median: 0, *p* < 0.01, Figure 1b).

The two groups of patients were comparable in terms of NEWScore 2 (Dept1 median: 5, Dept2 median: 4, Figure 1c, Table 1), and patients whose aggregate NEWScore 2 was 7 could remain in the internal medicine department if they did not require intensive respiratory support. There was no significant difference in the concentrations and distribution of 25(OH)D at the time of admission (Dept1: mean 29 ng/mL and Dept2: mean 29 ng/mL) (Table 1, Figure 1a). Significantly fewer patients died in Dept1, where all patients received a moderately high dose of vitamin D3 supplementation, compared to Dept2, where vitamin D3 supplementation was not included in the treatment protocol (10 vs. 29, respectively, *p* < 0.01, Table 1; *p* < 0.01, Figure 1d). Thus, mortality reduced by 67% (from 40 to 13%) in supplemented patients. This difference was detectable to almost the same extent in the vitamin D-deficient and -non-deficient groups (Table 1, deficient: 14 vs. 5 and non-deficient: 15 vs. 5).

### 3.2. Subgroup Analysis of the 30 Randomly Selected Patients in Dept1

In this part of this study, 30 out of the 76 patients supplemented in Dept1 were randomly selected from the previous analysis, and their concentrations of 25(OH)D and 1,25(OH)_2_D were measured on days 0, 4, and 8. All patients received vitamin D3 supplementation with doses of either 12,000 or 30,000 IU to observe the rate of increase in 25(OH)D and 1,25(OH)_2_D concentrations. After moderately high-dose supplementations, patients in Dept1 were supplemented with a standard dose of 3000 IU/day of vitamin D3. The characteristics of this patient subgroup are summarized in Table 2, and in Appendix A Table A2, chronic diseases are also listed.

We found that patients with vitamin D deficiency at admission (day 0) had significantly increased concentrations of 25(OH)D after supplementation on days 4 and 8 (day 4: *p* < 0.01, day 8: *p* < 0.01; Figure 2a). Additionally, in these vitamin D-deficient patients, the concentration of 1,25(OH)_2_D also significantly increased during the moderately high-dose supplementation period, remaining within but not exceeding the normal reference range of 18–78 pg/mL (day 4: *p* = 0.04; Figure 2a, Appendix A Table A3). 1,25(OH)_2_D concentrations did not further increase during the 3000 IU/day supplementation, and they did not significantly decrease or fall below the concentrations measured on day 0. Table A3 illustrates the increase in 1,25(OH)_2_D concentrations on days 4 and 8 for patients receiving 12,000 IU of vitamin D3 for 7 days and 30,000 IU of vitamin D3 for 3 days. In addition, 25(OH)D concentrations on day 8 significantly increased in non-deficient patients as well after supplementation (*p* = 0.04); however, they did not affect the 1,25(OH)D concentrations.

The average value of 25(OH)D in vitamin D-deficient patients increased from 11 ng/mL (day 0) to 27 ng/mL (day 8), while the increase was much smaller in non-deficient patients, from 37 ng/mL (day 0) to 45 ng/mL (day 8) (Table 3). The 25(OH)D concentration remained in the normal range in both cases, and the safety was also supported by the fact that the calcium and phosphate values did not rise above the normal range either (calcium concentration in vitamin D-deficient group on day 8 was 2.2 mmol/L and that in non-deficient group on day 8 was 2.0 mmol/L; phosphate concentration in vitamin D-deficient and -non-deficient groups on day 8 was 1.1 mmol/L) (Table 3).

## 4. Discussion

Our two-center, retrospective study is, to the best of our knowledge, the first clinical investigation examining the effects of short, moderately high-dose vitamin D3 supplementation on mortality in COVID-19 infection. We compared patients infected with SARS-CoV-2 virus across two clinical departments, assessing disease severity using the NEWScore 2. The concentrations of 25(OH)D measured at admission and the clinical protocols implemented—excluding vitamin D3 supplementation—were consistent between the two departments. Additionally, we evaluated vitamin D3 supply and the concentrations of 25(OH)D and 1,25(OH)_2_D during the supplementation. Our findings indicate that low 25(OH)D concentrations had no effect, but the moderately high-dose vitamin D3 supplementation significantly reduced mortality among hospitalized patients with COVID-19 infection. The vitamin D3 replacement protocol we implemented notably elevated the concentrations of 25(OH)D and 1,25(OH)_2_D in patients with vitamin D deficiency. This increase was less pronounced in patients who were not vitamin D-deficient, with no cases exceeding normal serum concentrations. The safety of this supplementation was further supported by the observation that serum calcium and phosphate concentrations remained within normal ranges.

Despite existing contradictions, numerous publications have highlighted the positive effects of vitamin D on the immune system and its role in the prevention of infection [29,30]. 1,25(OH)_2_D is known to decrease the activity of Th1/Th17 CD4+ T cells and related cytokine concentrations, increase regulatory T cells, and inhibit dendritic cell differentiation [31]. Meta-analyses have demonstrated that adequate vitamin D concentrations are associated with a reduced incidence of respiratory tract infections [32]. The COVID-19 pandemic, the most significant epidemic of the last decade, has profoundly impacted all components of the respiratory system, including the neuromuscular respiratory system, conducting airways, alveoli, pulmonary vascular endothelium, and pulmonary blood flow [33,34,35]. Both innate and adaptive immune responses are crucial for defense against COVID-19 infection [36]. Furthermore, vitamin D appears to play a role not only in combating COVID-19 but also in influencing the severity of the disease, as supported by the following studies.

Nielsen NM et al., observational clinical study examined the relationship between the severity of COVID-19 infection and concentrations of 25(OH)D. Data from 447 patients with a positive SARS-CoV-2 test were included in their study. Low concentrations of 25(OH)D were associated with a higher risk of severe COVID-19 infection [37]. Similarly, in our study, the in-hospital mortality was significantly higher among those who were vitamin D-deficient and did not receive vitamin D3 supplementation. In addition, in-hospital mortality was also significantly higher in non-deficient patients when they did not receive supplementation. Our results also show that patients, regardless of vitamin D status at admission, had significantly better survival if they were subsequently supplemented with moderately high-dose vitamin D3 supplementation.

In the study by Annweiler C. et al., the relationship between vitamin D3 supplementation and both the mortality and severity of COVID-19 infection was examined. They investigated whether a single dose of 80,000 IU of vitamin D3 supplementation could improve the severity and survival of patients living in nursing homes with COVID-19 infection. The primary outcome was mortality among those infected with COVID-19, while the secondary outcome assessed clinical severity using the Organization’s Ordinal Scale for Clinical Improvement (OSCI) score. Their results indicated that 82.5% of participants in the intervention group survived the COVID-19 infection, compared to only 44% in the control group. Furthermore, vitamin D3 supplementation was inversely related to the OSCI score, suggesting that vitamin D3 supplementation in this population was associated with less severe COVID-19 infections and better survival rates [38]. In our study, mortality was also significantly lower in patients receiving vitamin D3 supplementation at a similar total dose but administered in divided doses of vitamin D3. Additionally, we confirmed this association in a larger cohort of hospitalized patients with a matched control group while also measuring vitamin D concentrations. 25(OH)D and 1,25(OH)2D concentration measurements, which could have confirmed both safety and efficacy, were not performed in their study.

Similar to the previous study, Murai and colleagues studied 237 hospitalized patients with moderate to severe COVID-19 infection. The purpose of this multicenter, double-blind, randomized, placebo-controlled study was to determine whether a single high-dose vitamin D3 supplement (200,000 IU) affects the duration of hospital stay, with secondary outcomes including mortality, the proportion of patients admitted to the ICU, the number of patients who needed mechanical ventilation, and the duration of mechanical ventilation. A total of 117 patients was assigned to the vitamin D3 group, and there was no significant difference between the initial mean 25(OH)D concentrations of the two groups. It took approximately 11 days from the onset of symptoms to vitamin D3 supplementation. The results indicated that the length of hospital stay was not significantly reduced, the median length of stay for the vitamin D3 group was 7 days, and the placebo group also had a median stay of 7 days. Furthermore, there was no significant difference in mortality between the two groups. Additionally, there were no significant differences in ICU admission rates or the need for mechanical ventilation and in the mean duration of mechanical ventilation. Importantly, the single dose of vitamin D3 significantly increased baseline 25(OH)D concentrations, resulting in 91 patients achieving concentrations above 30 ng/mL [39]. Our study differed in several respects; in ours, the severity of illness at the time of hospitalization did not differ between the vitamin D3-supplemented and -non-supplemented groups, and treatment for COVID-19 followed the same protocol except for vitamin D3 supplementation. We administered a lower total dose of vitamin D3 spread over several days. However, our study was not placebo-controlled, double-blind, or randomized, which undoubtedly weakens the validity of our findings. Furthermore, a significant difference is that, in our study, mortality decreased in the supplemented group after the initial moderately high dose of vitamin D3. The benefits of continuous supplementation compared to a single high-dose bolus of cholecalciferol and its biological effect are also supported by the study by Griffin et al., which indicates that while bolus vitamin D3 does not significantly reduce the risk of respiratory infections, daily supplementation does. The biological explanation for this phenomenon involves the increased activity of the long-half-life 24-hydroxylase, which is responsible for the inactivation of vitamin D. This can paradoxically lead to intracellular calcitriol deficiency, thereby reducing the activity of immune cells. Additionally, the rise in FGF23 following a vitamin D3 bolus, which persists long after administration, significantly suppresses the 1α-hydroxylation of 25(OH)D. This process reduces the intracellular activation of vitamin D to 1,25(OH)_2_D, further contributing to diminished immune defense capacity [40]. Furthermore, cathelicidin LL-37, a vitamin D-inducible antimicrobial peptide, inhibits the binding of SARS-CoV2 to host cells, suggesting that vitamin D3 supplementation given as soon as possible after the onset of symptoms may improve the outcome of COVID-19 infection [41]. Consequently, the negative results of their study can be explained by the delayed vitamin D3 supplementation.

Ling and colleagues reported a multicenter retrospective study aiming to determine whether serum 25(OH)D concentrations influence mortality in COVID-19 patients and to assess the effect of very different doses of vitamin D3 supplementation (from 20,000 IU/every 2 weeks to 40,000 IU/daily for 7 days). The study included 444 participants, of whom 309 had 25(OH)D concentrations lower than the normal range. According to the data, no significant association was found between initial 25(OH)D concentrations or vitamin D deficiency and COVID-19 mortality when analyzing all patients. However, cholecalciferol supplementation was associated with improved survival outcomes, regardless of baseline 25(OH)D concentrations [42]. Our study showed similar results in several respects. We also found no association between initial vitamin D concentration and mortality. Specifically, the mortality rate for patients with vitamin D deficiency at hospital admission was 19%, compared to 21% for those without vitamin D deficiency in the non-supplemented group. However, vitamin D3 supplementation significantly reduced mortality, regardless of initial vitamin D levels. Notably, our study was able to establish this overall association with a uniform vitamin D3 dose by comparing two similar groups.

Our study had certain limitations, including its single-center design, retrospective and non-randomized nature, and the relatively small sample size that may impact representativeness. Additionally, we did not measure urinary calcium levels when assessing vitamin D safety, a gap attributed to infection control protocols in place during the epidemic. However, a strength of our study is that it was conducted in two departments of the same clinic, where the dosage of vitamin D3 received by patients in Dept1 was uniform, and the patient care protocols were similar in both departments, except for the introduction of vitamin D3 supplementation in Dept2 after the completion of our study. Furthermore, all variables were measured in the same central laboratory, and patients were selected randomly, with the NEWScore 2 calculated at admission showing no significant differences. Finally, the concentration of active vitamin D (1,25(OH)_2_D) was measured concurrently with 25(OH)D. Considering the weaknesses and strengths of our study, a large multicenter, randomized controlled trial would be needed to confirm our results.

## 5. Conclusions

Our results demonstrate that in two COVID-19-infected hospitalized patient populations with similar characteristics treated with the same protocol—except for vitamin D3 supplementation—moderately high-dose vitamin D3 supplementation reduces mortality, an effect not limited to patients with vitamin D deficiency. This vitamin D3 supplementation is both safe and effective in the treatment of COVID-19 infection. Based on our data, we recommend the introduction of 3 × 30,000 IU of vitamin D3 in hospitalized patients infected with COVID-19 as it is safe and has a positive impact on mortality.

## Figures and Tables

**Figure 1 nutrients-17-00507-f001:**
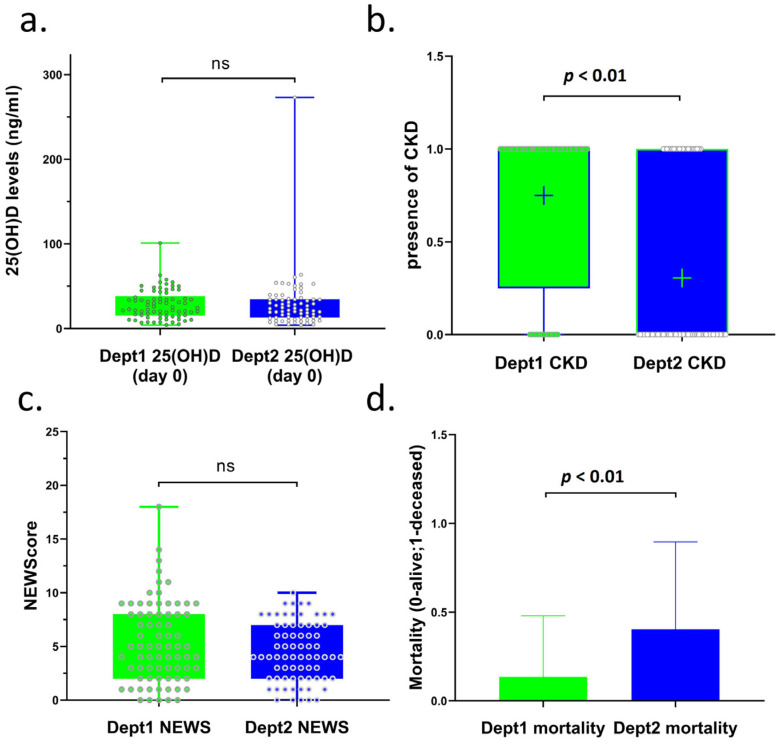
Comparison of 25(OH)D concentrations, CKD stages, NEWScore 2, and mortality of COVID-19 patients in two departments of our clinic (Dept1 and Dept2). (**a**) 25(OH)D concentrations of COVID-19 patients in Dept1 (mean: 29) and Dept2 (mean: 29). (**b**) Presence of CDK in COVID-19 patients in Dept1 (median: 1) and Dept2 (median: 0). (**c**) NEWScore 2 of COVID-19 patients in Dept1 (median: 5) and Dept2 (median: 4). (**d**) Mortality of COVID-19 patients in Dept1 (mean: 10/76) and Dept2 (mean: 29/72). Interpretation of the figures: ns: not significant; significant if *p* < 0.05; (**a**) box—25–75 percentile; whiskers—min to max; dot—all patients; (**b**) mean signed as +; box—25–75 percentile; whiskers—min to max; (**c**) box—25–75 percentile; whiskers—min to max; dots—all patients; (**d**) box—mean; whiskers—standard deviation (SD); CKD: chronic kidney disease.

**Figure 2 nutrients-17-00507-f002:**
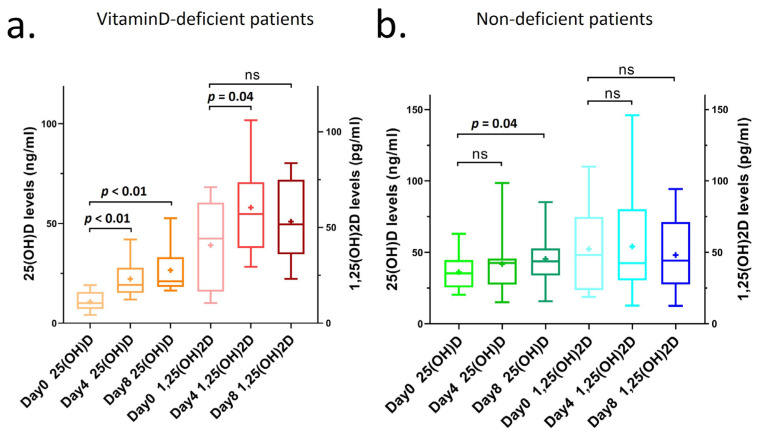
25(OH)D and 1,25(OH)_2_D concentrations of COVID-19 patients with supplementation in Dept1 on days 0, 4, and 8 after hospitalization. (**a**) 25(OH)D and 1,25(OH)_2_D concentrations of COVID-19 patients with vitamin D deficiency (<20 ng/mL at day 0) and after supplementation in Dept1 on days 0, 4, and 8 during hospitalization. (**b**) 25(OH)D and 1,25(OH)_2_D concentrations of COVID-19 patients with non-deficient vitamin D concentrations (≥20 ng/mL at day 0) and after supplementation in Dept1 on days 0, 4, and 8 during hospitalization. Interpretation of the figures: ns: not significant; significant if *p* < 0.05; line: median; mean signed as +; box: 25–75 percentile; whiskers: min to max.

**Table 1 nutrients-17-00507-t001:** Clinical variables and NEWScore 2 of the 148 patients included in this study. NEWScore 2: Aggregate score of 0–4 means low clinical risk and a Ward-based response, 5–6 means medium risk and a key threshold for an urgent response, and 7 or more indicates high clinical risk and an urgent or emergency response. However, patients with this latter score in this table did not require mechanical ventilation, allowing for their treatment in our departments rather than in the ICU. Deficient: (<20 ng/mL), non-deficient: (≥20 ng/mL); significance level highlighted in bold text: *p* < 0.05, a: not significant calculated with chi-squared test, SD: standard deviation, ng/mL: concentration measure, N (%): number of the variables and it’s percent ratio.

	Dept1 (Patients Received Vitamin D3 Supplementation)	Dept2 (Patients Did Not Receive Vitamin D3 Supplementation)	*p*-Value
**Gender** (All) **(N (%))**	76 (100%)	72 (100%)	
Male **(N (%))**	43 (57%)	39 (54%)	
Female **(N (%))**	33 (43%)	33 (46%)	
**Mean Age (Year (±SD))**	69 (±16)	66 (±14)	
**NEWScore 2 at the hospitalization**	**Patients in Dept1**	**Patients in Dept2**	***p*-value**
Aggregate score 0–4 (low risk) (N (%))	36 (47%)	39 (54%)	0.41
Aggregate score 5–6 (medium risk) (N (%))	12 (16%)	12 (17%)	0.88
Aggregate score 7 or more (high risk) (N (%))	28 (37%)	21 (29%)	0.32
**Patients at hospital admission**			
25(OH)D-deficient (<20 ng/mL) (all) (N (%))	24 (32%)	30 (42%)	0.20 a
25(OH)D-non-deficient (≥20 ng/mL) (all) (N (%))	52 (68%)	42 (58%)	
25(OH)D-deficient died in hospital (N (%))	5 (6.6%)	14 (19%)	**0.048**
25(OH)D-non-deficient died in hospital (N (%))	5 (6.6%)	15 (21%)	**<0.01**
**25(OH)D concentration at hospital admission**			
All (mean ± **SD) (ng/mL)**	29 (±17)	29 (±33)	not significant
25(OH)D-deficient (<20 ng/mL) (mean ± **SD) (ng/mL)**	11 (±4.6)	12 (±4.7)	not significant
25(OH)D-non-deficient (≥20 ng/mL) (mean ± **SD) (ng/mL)**	37 (±15)	41 (±38)	not significant

**Table 2 nutrients-17-00507-t002:** Characteristics of the 30 randomly selected vitamin D3-supplemented patients in Dept1. SD: standard deviation.

**Gender (All)**	**30 Patients (100%)**
Male	18 patients (60%)
Female	12 patients (40%)
**Mean age (years (±SD))**	70 (±17)
**Average number of days spent in hospital (±SD)**	13 (±5.4)
**Vitamin D3 supplementation**	
25(OH)D dose—12,000 IU (7 days)	9 patients (30%)
25(OH)D dose—30,000 IU (3 days)	21 patients (70%)

**Table 3 nutrients-17-00507-t003:** Safety measures of 30 vitamin D3-supplemented patients included in this subgroup study. It includes the 25(OH)D, calcium, and phosphate concentrations on days 0, 4, and 8 (mean ± SD). ng/mL and mmol/L: concentrations, NA: not applicable.

Safety Measures of Vitamin D3 Supplementation	Day 0 Mean (±SD)	Day 4 Mean (±SD)	Day 8 Mean (±SD)
**25(OH)D** concentration **(ng/mL)**			
All	29 (±16)	37 (±20)	39 (±18)
Deficient (<20 ng/mL) at hospitalization	11 (±4.7)	22 (±10)	27 (±12)
Non-deficient (≥20 ng/mL) at hospitalization	37 (±13)	44 (±20)	45 (±17)
**Ca** concentration **(mmol/L)**			
All	2.1 (±0.1)	NA	2.1 (±0.3)
25(OH)D-deficient (<20 ng/mL) at hospitalization	2.1 (±0.1)	NA	2.2 (±0.2)
25(OH)D-non-deficient (≥20 ng/mL) at hospitalization	2.1 (±0.2)	NA	2.0 (±0.3)
**Phosphate** concentration **(mmol/L)**			
All	1.0 (±0.2)	NA	1.1 (±0.3)
25(OH)D-deficient (<20 ng/mL) at hospitalization	0.9 (±0.3)	NA	1.1 (±0.3)
25(OH)D-non-deficient (≥20 ng/mL) at hospitalization	1.1 (±0.2)	NA	1.1 (±0.3)

## Data Availability

The original data are not available because of ethical considerations.

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
