# Peer review of "Effect of Moderately High-Dose Vitamin D3 Supplementation on Mortality in Patients Hospitalized for COVID-19 Infection"

_nutrients, 2025, doi:10.3390/nu17030507_

Round 1
Reviewer 1 Report
Comments and Suggestions for Authors
Review for Nutrients:- MS- nutrients-3394942
Effect of moderate high dose vitamin D3 supplementation on mortality in patients hospitalised for COVID-19 infection by Sumegi et al.
This MS reports a retrospective observational study of mortality rates in two groups of patients admitted with equally severe COVID-19 illness and similar mean serum 25(OH)Ds on admission. Deficiency [25(OH)D<20ng/ml] was present in 54/148 patients overall and deficiency rates in the 2 groups were not significantly different. One group was treated routinely with 90,000 IU of vitamin D3 [given as 12,000 IU/day for 7 days or as 30,000 IU/day for 3 days] followed by 3000 IU/day thereafter and one group was not given vitamin D3. All patients received standardised best available treatment. Mortality in the unsupplemented group was similar in those +/- deficiency on admission [at 19.4% and 20.8% respectively]. 46.7% of unsupplemented patients died. In the supplemented group overall mortality was 6.6%, and did not differ between those with or without baseline deficiency.
In a subgroup study of vitamin D safety in 30 treated patients, deficiency was corrected within 4 days without inducing abnormal serum calcium or phosphate values or raising serum 25(OH)D or 1,25(OH)2D concentrations above normal and the increases in serum 25(OH)D were significantly greater in those with deficiency than in those without.
The report is clearly written and easy to follow in most sections, though some editing of the English language usage throughout the text would make it easier to follow at a first reading. Also, there are some features of the layout of Figures 1 and 2 that I had to look at more than once to understand. [e.g., see general comment 7]
General and Major comments.
1.The use of vitamin D in this study was as supplementation though some would call it treatment. Both terms are used in this MS and it would readers if you kept to one of these terms throughout.
2.’ca’ is a valid abbreviation, but little used in VitD reports; you might like to use ‘circa’ or ‘about’ instead?
3.You report serum 25(OH)D levels, though lab measurements are always of concentrations; it would be useful, therefore, to use ‘concentrations’ rather than ‘levels’ in sections often read in isolation, such as the abstract and conclusions.
4.For lab measurements where the findings are critical to the analyses made but are also known to have high variability, such as the vitamin D metabolite assays. it is customary to report the CVs recorded by the labs doing those assays and the QC scheme used by the relevant Lab is also reported. These data provide valuable reassurance about the accuracy of the data used in analysis and should be placed in the Methods section. The lab that did the immunoassays could, I am sure, easily provide that information as it is routinely included in each batch of immunoassays.
5.The term ‘parameters’ is used to describe the data reported on the clinical and laboratory features of the patients in this study. However, the term to use is ‘variables, since a ‘parameter’ is a specific value of a variable that has been chosen for use as a definition, e.g., to define vitamin D deficiency from serum 25(OH)D values.
6.The text on page 5 has an odd layout in the heading of Table 2. I think it was upset by the indentation of the text in the paragraph above the heading to Table 2. Please check and clarify.
7.Figures 1 (a and b) and 2 (a and b) took me several readings before I could understand them. The data shown would be easier to follow if shown in darker print. Also, the significance of the rows of ****’s was unclear to me until I got to reading the legends. Readers might, therefore, find this information easier to follow at a first reading if the actual p values were shown in each graph rather than rows of *s. One reason for this last difficulty was that no data is shown for Group 1 patients in Fig 1, b, c, or d, see re Fig 1 below.
8.the discussion section is rather long which is likely to inhibit readers from following all the points made. The text could be usefully cut back by some editing of the English usage without losing any information. If no native English speaker with a suitable scientific background is available to do this I would be willing to do this job.
Specific and editing comments, by section and line number.
Background. Line 16, please insert the word supplementation after vitamin D to make it clear what the study looked at. Line 27, changing ‘or’ between 25(OH)D and 1,25(OH)2D would read better. Line 29, ‘….supplemental dosage of …’ would read better than ‘…the applied dose…’
Introduction. Line 35. why not insert ‘calcitriol’ after the word D-hormone so that readers could look it up if necessary. Line 43, change ‘number of evidence’ to ‘amount of evidence’ for clarity. line 46, suggested not suggests [as that discovery is quite old]. Line 51, if you want to mention that part of the causation of cytokine storms you should reference the reduction of that particular factor by vitamin D. Line 60, ‘ …a daily supplement of…, not ‘supplementation of…’ line 70, if you want to mention why large bolus doses of vitamin D fail in COVID-19 you might like to quote the work showing they cannot prevent rickets, as mentioned in Griffin, Hewison et al. in their paper on COVID-19. The mechanisms causing very large bolus failure are known and are, basically, the mechanisms protecting people from toxicity when vitamin D provision is excessive. Lines 74-75 are hard to follow and need editing.
Materials and Methods. ….. line 92, do you mean that the non-supplemented group of patients started vitamin D at 3000 IU/day once the study had been completed? Line 102, again, as also in line 110, the ‘ ….3000IU/day for the remainder of the patients stay’, I take it that this applies to the treated group although, as in Line 92, this could apply to both groups – please clarify. Line 121. A total of 148 patients were …. Line 129, the amount of vitamin D given was …. Line 135, the clinical features of COVID-19…. Line 149 is where to put the data on the assay CVs and to put the name of the QC system in use. Line 158, ..patients supplemented [all were treated but only group 1 was supplemented] would be more clear for readers.
Results. Line 172, …. or not receiving….’ would be more clear. Line 178, ….was the higher incidence of CKD in Group 1 patients.
Line 184, if a NEWScore2 of 7 or higher requires an ‘urgent response’ but, from the earlier text
those needing intensive care went straight to such care, and were not included in the study,
what score triggered admission to ICU, this might go into Table 1 with 0 patients in both groups
if I have understood the text correctly. line 192/3 is an example of how the text could be reduced.
‘….. thus mortality was reduced by 67% [from 40.2% to 13.2%] in supplemented patients.’
Table 2 has lost the headings for the two patient groups.
Figure 1, see General comment 7, No data is shown in Fig 1, b, c or d. Even if not
Significantly from group 2 data it should be shown so that one can understand the Figure.
Line 222, were ‘all patients, put on 3000 IU/day’ which would include Group 2 patients – please
clarify. Line 226 Patients ‘were treated in Group 1, for COVID-19, but you mean were
supplemented with vitamin D, which should be mentioned in the heading; please clarify the text.
Figure 2 a and b, as for Figure 1 b-d, showing the actual data for both groups would make it
possible to see what is meant here at a first reading whereas it took me several re-reads.
Discussion. This section could be shortened without loss of content by efficient editing of
the English language, as mentioned earlier.
Line 148+ The studies discussed are relevant but reports on vitamin D in COVID-19 illness are still appearing in the literature; for example, a recent review and meta-analysis reported benefits of supplementation from 21 trials and 8 observational studies and showed mortality reductions in the observational but not the RCT studies.* It would be useful, therefore, to recheck the literature for possible reports of similar vitamin D supplementation to yours before finalising this MS. Line 347 reads oddly; group 2 did not get vitamin D but you say ‘the dose of vitamin D was the same’ when reporting that the protocol of care was the same for both patient groups and you then end that comment by saying ‘except for vitamin D supplementation, …? Please clarify that section of the text.
*Zhang X, Wu J, Dong H, Shang N, Li Y, Zhang Y, Guo S, Mei X. The impact of supplementing vitamin D through different methods on the prognosis of COVID-19 patients: a systematic review and meta-analysis. Front Nutr. 2024 Sep 25;11:1441847. doi: 10.3389/fnut.2024.1441847..
Comments on the Quality of English Languagesome comments and descriptions of the study are difficult, and a very few are impossible, to follow due to the English usage. Editing of the English usage, after the authors have clarified the matters I could not follow, would be helpful to readers. It would also be possible to shorten the rather overlong/wordy Discussion by editing the English usage
Author Response
Authors’ reply to Reviewer 1:
Thank you for your thorough work with helpful suggestions and notes to improve our manuscript.
We corrected the manuscript based on your suggestions and recommendations. The corrections are seen in the revised manuscript and below as green text.
Please see below the point by point answers to your suggestions:
“Review for Nutrients: -MS- nutrients-3394942
Effect of moderate high dose vitamin D3 supplementation on mortality in patients hospitalised for COVID-19 infection by Sumegi et al.
This MS reports a retrospective observational study of mortality rates in two groups of patients admitted with equally severe COVID-19 illness and similar mean serum 25(OH)Ds on admission. Deficiency [25(OH)D<20 ng/mL] was present in 54/148 patients overall and deficiency rates in the 2 groups were not significantly different. One group was treated routinely with 90 000 IU of vitamin D3 [given as 12 000 IU/day for 7 days or as 30 000 IU/day for 3 days] followed by 3000 IU/day thereafter and one group was not given vitamin D3. All patients received standardised best available treatment. Mortality in the unsupplemented group was similar in those +/- deficiency on admission [at 19.4% and 20.8% respectively]. 46.7% of unsupplemented patients died. In the supplemented group overall mortality was 6.6%, and did not differ between those with or without baseline deficiency.
In a subgroup study of vitamin D safety in 30 treated patients, deficiency was corrected within 4 days without inducing abnormal serum calcium or phosphate values or raising serum 25(OH)D or 1,25(OH)2 D concentrations above normal and the increases in serum 25(OH)D were significantly greater in those with deficiency than in those without.
The report is clearly written and easy to follow in most sections, though some editing of the English language usage throughout the text would make it easier to follow at a first reading. Also, there are some features of the layout of Figures 1 and 2 that I had to look at more than once to understand. [e.g., see general comment 7]
General and Major comments.
1.The use of vitamin D in this study was as supplementation though some would call it treatment. Both terms are used in this MS and it would readers if you kept to one of these terms throughout.
We corrected this based on your suggestion and change the “treatment” to “supplementation” in the manuscript referring to vitamin D supplementation. See in lines 102,153,217,222,302,338,385,447
2.’ca’ is a valid abbreviation, but little used in VitD reports; you might like to use ‘circa’ or ‘about’ instead?
Thanks for the comment, we corrected the abbreviation ‘ca’ to circa. See in lines 26,104
3.You report serum 25(OH)D levels, though lab measurements are always of concentrations; it would be useful, therefore, to use ‘concentrations’ rather than ‘levels’ in sections often read in isolation, such as the abstract and conclusions.
We corrected the “levels” to “concentration”, although the levels is regularly used in manuscripts and the daily routine as well. See in lines 28,30,54,56,61,72,93,99,101,102,106,110,124,134 and we corrected all incorrect words.
4.For lab measurements where the findings are critical to the analyses made but are also known to have high variability, such as the vitamin D metabolite assays. it is customary to report the CVs recorded by the labs doing those assays and the QC scheme used by the relevant Lab is also reported. These data provide valuable reassurance about the accuracy of the data used in analysis and should be placed in the Methods section. The lab that did the immunoassays could, I am sure, easily provide that information as it is routinely included in each batch of immunoassays.
We added these QC data requested from the lab into the methods section as seen in line 146-148.
5.The term ‘parameters’ is used to describe the data reported on the clinical and laboratory features of the patients in this study. However, the term to use is ‘variables, since a ‘parameter’ is a specific value of a variable that has been chosen for use as a definition, e.g., to define vitamin D deficiency from serum 25(OH)D values.
We corrected the ‘parameter’ to ‘variables’ as seen in lines 100,138,182,377,409.
6.The text on page 5 has an odd layout in the heading of Table 2. I think it was upset by the indentation of the text in the paragraph above the heading to Table 2. Please check and clarify.
Thank you for your notice. However, not the heading had been lost but the patient number was incorrectly indicated. Only the subgroup of Dept1 patients, supplemented 30 patients were used for safety evaluation, where 1,25(OH)D measurements also were carried out. We corrected the wrong patient number to 30 as seen in line 227. Furthermore, the 25(OH)D concentration for the entire patient group was included in Table 1 (from line 182 to 189).
7.Figures 1 (a and b) and 2 (a and b) took me several readings before I could understand them. The data shown would be easier to follow if shown in darker print. Also, the significance of the rows of ****’s was unclear to me until I got to reading the legends. Readers might, therefore, find this information easier to follow at a first reading if the actual p values were shown in each graph rather than rows of *s. One reason for this last difficulty was that no data is shown for Group 1 patients in Fig 1, b, c, or d, see re Fig 1 below.
We removed the asterisks from the figures and added the p value given by the software we calculated, also we bold the text to be more visible. We also add to the figure legend ns: not significant. Additionally, data into the legend in Figure 1 b, c, and d. as these can be seen from line 203-214 and 241-250.
8.the discussion section is rather long which is likely to inhibit readers from following all the points made. The text could be usefully cut back by some editing of the English usage without losing any information. If no native English speaker with a suitable scientific background is available to do this I would be willing to do this job.
We modified the discussion section in a significant extent as the other reviewer also suggested to extend with additional references. Thus, we added extra references and cut back the text where this was important as distracted the flow of information. If the manuscript will be accepted, we would like to submit it to the journal for English proofreading
Specific and editing comments, by section and line number.
Background.
Line 16, please insert the word supplementation after vitamin D to make it clear what the study looked at.
We corrected the requested changes in line 16 as seen in line 18-19.
Line 27, changing ‘or’ between 25(OH)D and 1,25(OH)2D would read better.
Thank you, we corrected the requested change in line 27. See in line 30.
Line 29, ‘….supplemental dosage of …’ would read better than ‘…the applied dose…’
Yes, we agree with the suggestion, we have entered the term ‘supplemetal dosage’ of instead of ‘applied dose’. See in line 33.
Introduction.
Line 35. why not insert ‘calcitriol’ after the word D-hormone so that readers could look it up if necessary.
Thank you, we agree with the comment, but we deleted this sentence due to the changes of our manuscript.
Line 43, change ‘number of evidence’ to ‘amount of evidence’ for clarity.
Thank you for your comment, we have changed the term number of evidence to amount of evidence. See in line 38.
Line 46, suggested not suggests [as that discovery is quite old].
Thank you, we agree, the correct expression is ‘suggested’, which we have corrected in the manuscript. See in line 41.
Line 51, if you want to mention that part of the causation of cytokine storms you should reference the reduction of that particular factor by vitamin D.
Thank you for your comment, we have added a new reference describing the effect of vitamin D on individual cytokines. Based on this, we have supplemented the discussion section of manuscript. See in line 45-46.
Line 60, ‘ …a daily supplement of…, not ‘supplementation of…’
We corrected the requested changes in line 60 as seen in line 55 now.
Line 70, if you want to mention why large bolus doses of vitamin D fail in COVID-19 you might like to quote the work showing they cannot prevent rickets, as mentioned in Griffin, Hewison et al. in their paper on COVID-19. The mechanisms causing very large bolus failure are known and are, basically, the mechanisms protecting people from toxicity when vitamin D provision is excessive.
We added the reference and a sentences with explanation as you suggested for line 70 as seen now in line 343-353, but eventually due to changes this reference and explanation was moved to the discussion section.
Lines 74-75 are hard to follow and need editing.
Due to requested changes to the manuscript, this section was deleted rather than modified, and the text was reworded instead.
Materials and Methods.
Line 92, do you mean that the non-supplemented group of patients started vitamin D at 3000 IU/day once the study had been completed?
Yes, we did as you wrote, however it was the same as in the Dept1 mentioned in the manuscript, not only 3000IU, this was just the maintenance dose. We corrected the sentence to be clear this information as seen in line 86-89.
Line 102, again, as also in line 110, the ‘ ….3000IU/day for the remainder of the patients stay’, I take it that this applies to the treated group although, as in Line 92, this could apply to both groups – please clarify.
We cleared the meaning in the “line 92” seen now in line 86-89 as we described above. Also, we reformatted this misunderstanding sentence to have clear information as seen in line 107-108. We changed the ‘remainder’ to ‘during’.
Line 121. A total of 148 patients were ….
Thank you for your comment, we corrected it as requested. See in line 118.
Line 129, the amount of vitamin D given was ….
Thank you for your comment, we corrected it as requested. See in line 127.
Line 135, the clinical features of COVID-19….
We agree with the comment and corrected it in the manuscript. See in line 132.
Line 149 is where to put the data on the assay CVs and to put the name of the QC system in use.
We agree with the proposal and we inserted the requested information into the manuscript. See in line 146-148.
Line 158, ..patients supplemented [all were treated but only group 1 was supplemented] would be more clear for readers.
Thank you, we agreed, we wrote ‘supplemented’ instead of ‘treated’. It can be seen in green text in 14 places from lines 102,152,159,199,217,222,227,262,302,338 (2 pieces), 343,365,447.
Results.
Line 172, …. or not receiving….’ would be more clear.
Thank you, we agree, we have corrected the manuscript. See in line 172.
Line 178, ….was the higher incidence of CKD in Group 1 patients.
Thank you for the suggestion, we have corrected the manuscript. See in line 179.
Line 184, if a NEWScore2 of 7 or higher requires an ‘urgent response’ but, from the earlier text those needing intensive care went straight to such care, and were not included in the study,what score triggered admission to ICU, this might go into Table 1 with 0 patients in both groups if I have understood the text correctly.
Thank you for your notice. We cleared out the possibility of the misunderstanding with a sentence added to the legend of Table 1.: “However, patients with this score in this table did not require mechanical ventilation, thus could be treated in our departments instead of ICU.” as seen in line 185-187.
Line 192/3 is an example of how the text could be reduced.
‘….. thus mortality was reduced by 67% [from 40.2% to 13.2%] in supplemented patients.’
Thank you, we have rewritten the text according to the example. See in line 198-199.
Table 2 has lost the headings for the two patient groups.
Thank you for your notice. However, not the heading had been lost but the patient number was incorrectly indicated. Only the subgroup of Dept1 patients, supplemented 30 patients were used for safety evaluation, where 1,25(OH)D measurements also were carried out. We corrected the wrong patient number to 30 as seen in line 227. Furthermore, the 25(OH)D concentration for the entire patient group was included in Table 1 (between line 188-189.)
Figure 1, see General comment 7, No data is shown in Fig 1, b, c or d. Even if not significantly from group 2 data it should be shown so that one can understand the Figure.
In the figure we regularly present the distribution of the data with mean, median and SD or percentiles etc. We think adding additional data to the figures would make it busier and did not help to understand the meaning of them. Thus, we added extra information to the Figure 1 legend with the meaning of the labelling and also in (b) medians, in (c) means as well as in (d) ratios, which original data can be found both in in Table 1 and the relevant texts. We also added extra word referencing the Table 1 and Figure 1 if were missing in the text. The corrections can be seen from line 203 to 214.
Line 222, were ‘all patients, put on 3000 IU/day’ which would include Group 2 patients – please clarify.
We corrected the sentence to be clear the information only Dept1 were supplemented with standard does as well as it can be seen in line 221-222.
Line 226 Patients ‘were treated in Group 1, for COVID-19, but you mean were supplemented with vitamin D, which should be mentioned in the heading; please clarify the text.
We added the ’vitamin D3 supplemented’ in the heading instead of treatment as shown in line 227
Figure 2 a and b, as for Figure 1 b-d, showing the actual data for both groups would make it possible to see what is meant here at a first reading whereas it took me several re-reads.
We corrected the figures and changed the asterisk to the actual p values, and also added interpretation of ‘ns’ to be clear this is for not significant results. The figure legend has the line and plus sign explained and the Y axis had values which can be read. With all the original mean and median values will be too busy the figure, thus we did not include them. See in line 241-250.
Discussion.
This section could be shortened without loss of content by efficient editing of the English language, as mentioned earlier.
As the other reviewer also requested more references to the discussion, we add more information but the original part we shortened, reducing the repetitions and if the manuscript will be accepted, we would like to submit it to the journal for English proofreading
Line 148+ The studies discussed are relevant but reports on vitamin D in COVID-19 illness are still appearing in the literature; for example, a recent review and meta-analysis reported benefits of supplementation from 21 trials and 8 observational studies and showed mortality reductions in the observational but not the RCT studies.* It would be useful, therefore, to recheck the literature for possible reports of similar vitamin D supplementation to yours before finalising this MS.
Thank you, we added more references discussing the differences and similarities to our study as can be seen in line 62-68.
Line 347 reads oddly; group 2 did not get vitamin D but you say ‘the dose of vitamin D was the same’ when reporting that the protocol of care was the same for both patient groups and you then end that comment by saying ‘except for vitamin D supplementation, …? Please clarify that section of the text.
Thank you for your notice, we deleted that part with ‘same dose’ etc and cleared the meaning of the sentence by adding extra words at the end as can be seen now in line 373-377.
*Zhang X, Wu J, Dong H, Shang N, Li Y, Zhang Y, Guo S, Mei X. The impact of supplementing vitamin D through different methods on the prognosis of COVID-19 patients: a systematic review meta-analysis. Front Nutr. 2024 Sep 25;11:1441847. doi: 10.3389/fnut.2024.1441847..
Comments on the Quality of English Language
some comments and descriptions of the study are difficult, and a very few are impossible, to follow due to the English usage. Editing of the English usage, after the authors have clarified the matters I could not follow, would be helpful to readers. It would also be possible to shorten the rather overlong/wordy Discussion by editing the English usage
We have tried to improve the language quality of the article. In this way, we hope that readers will find the text easy to follow everywhere.
Once again, we would like to thank the reviewer for his thorough and comprehensive work. His comments and observations have greatly improved the quality of our article. We hope that you will find our article suitable for publication after our corrections.
Reviewer 2 Report
Comments and Suggestions for Authors
1. There is no need to devote too much space at the beginning to the role of vitamin D. This is not a major concern of the readers.
2. A very large number of studies have been conducted on COVID-19 and vitamin D. The literature review of this study is clearly inadequate.
3. This study does not provide a conclusion that is significantly different from the available evidence. Both the practical and theoretical implications of the conclusion are clearly insufficient to support the clinical decision-making process that is already in place.
4. The small sample size of the study is not sufficient to support the generalisation of the findings.
5. The manuscript continues to repeat the findings in the discussion section without discussing how the findings can be applied in practice and what are the novel and unique perspectives.
Author Response
Authors’ reply to Reviewer 2:
Thank you for your work with suggestions and notes to improve our manuscript. We corrected the manuscript based on your suggestions and recommendations. The corrections are seen in the revised manuscript and below as green text.
Please see below the point-by-point answers to your suggestions:
- There is no need to devote too much space at the beginning to the role of vitamin D. This is not a major concern of the readers.
We deleted some part of the information related to the role of vitamin D and the modified part can be seen in line 38-46.
- A very large number of studies have been conducted on COVID-19 and vitamin D. The literature review of this study is clearly inadequate.
We are agree that there are very large number of studies on COVID19 and vitamin D (around 623), however, few of them relevant to our study either of the outcomes indicate (from 295 only 11 discussed directly mortality, but the supplementation doses were different [form very low (600 IU) to very high (200 000 IU)doses] and the dosing times also differed. Retrospective study and similar number of patients (100-200 patients) were 16 and 19 studies. Also, some of them did not measured the increase of 25(OH)D levels after supplementation, only in 28 studies were the 25(OH)D concentration controlled and only 3 studies measured the 1,25(OH)D levels and other safety parameters, what we did and added adequate conclusion related to our findings.
Mortality was examined:
doi:10.1038/s41598-022-24053-4
doi:10.3390/nu15153470
doi:10.3390/nu15224818
doi:10.1007/s11606-021-07170-0
doi:10.5507/bp.2023.045
Control 25(OH)D measurement after supplementation:
doi:10.1136/bmj-2022-071245
doi:10.3390/jcm10112378
doi:10.1093/pubmed/fdae007
doi:10.1136/postgradmedj-2020-139065
doi:10.1016/j.clnu.2021.03.001
Studies with similar number of patients to our study:
doi:10.3390/nu14245204
doi:10.1186/s13063-023-07107-4
doi:10.1007/s12020-023-03481-w
doi:10.1002/iid3.844
doi:10.1080/07315724.2020.1856013
Studies in which 1,25(OH)2D was measured:
doi:10.1016/j.clnu.2021.03.001
doi:10.1016/j.arcmed.2023.02.002
doi:10.3390/diagnostics14131408
- This study does not provide a conclusion that is significantly different from the available evidence. Both the practical and theoretical implications of the conclusion are clearly insufficient to support the clinical decision-making process that is already in place.
Reading the 295 roughly related manuscript, our study is the only one, which added the same, well controlled moderate high dose vitamin D3 supplementation with a control group with the same treatment protocol except the supplementation, who investigated mortality and also measured the increase of 25(OH)D after supplementation as well as 1,25(OHD), and other safety measures such as Ca, P during the high dose supplementation period.
We have therefore added to our article, highlighting its specificity and underlining our proposal for a modification of the treatment protocol.
- The small sample size of the study is not sufficient to support the generalisation of the findings.
Our sample size is comparable to those original articles, even RCTs, which recently or earlier published, even in 2023 was published one in ’Nutrients’ with 78 supplemented and 77 control patients. /Ref: Domazet Bugarin, Josipa et al. “Vitamin D Supplementation and Clinical Outcomes in Severe COVID-19 Patients-Randomized Controlled Trial.” Nutrients vol. 15,5 1234. 28 Feb. 2023, doi:10.3390/nu15051234
- The manuscript continues to repeat the findings in the discussion section without discussing how the findings can be applied in practice and what are the novel and unique perspectives.
We reframe the discussion by adding extra references with more details but shortened back the repetitions as can be seen in the final version of the article.
Again, we thank the reviewer for his work, including for pointing out that we should better highlight the uniqueness of the article and its potential impact on clinical practice.

Reviewer 3 Report
Comments and Suggestions for Authors
Dear authors, the high level of similarities is not acceptable. The 34% similarity index is too high! First, you must rewrite your work before it can be considered for publication.
In the abstract, provide more data on your background. It Is not enough to mention that “the effects of vitamin D3 on the risk and complications of infections have not yet been fully elucidated”.
Line 21: “…two departments of our clinic…” – Which departments? Which clinic? This is too vague.
Lines 29-30: Elaborate on your Conclusions. What can the readers learn from your study and what is expected to be done next, based on your obtained results?
In the Introduction, would be great if you could provide more studies on Vitamin D and COVID-19 worldwide.
In section 2, you have to justify your sample size regarding representativeness. 148 patients from only one clinic, in my point of view, is an important limitation of your research, especially considering you are submitting this study to a Q1 international journal.
The Discussion can also be improved and more data from other studies conducted in other regions of the world should be mentioned.
A Conclusions section is missing. Future perspectives should also be provided.
Author Response
Authors’ reply to Reviewer 3
Thank you for your work with suggestions and notes to improve our manuscript. We corrected the manuscript based on your suggestions and recommendations. The corrections are seen in the revised manuscript and below as green text. Please see below the point by point answers to your suggestions:
„Comments and Suggestions for Authors
Dear authors, the high level of similarities is not acceptable. The 34% similarity index is too high! First, you must rewrite your work before it can be considered for publication.
We ran Turnitin, the result is 22%. If the manuscript will be accepted, we will also send it to the journal for English proofreading, which will probably help.
In the abstract, provide more data on your background. It Is not enough to mention that “the effects of vitamin D3 on the risk and complications of infections have not yet been fully elucidated”.
We reframe the background in the abstract added more information as you requeste, please see in line 18-20.
Line 21: “…two departments of our clinic…” – Which departments? Which clinic? This is too vague.
We have specified the departements and clinic in line 23-24.
Lines 29-30: Elaborate on your Conclusions. What can the readers learn from your study and what is expected to be done next, based on your obtained results?
We added more specific conclusion with future perspective as can be seen in line 33-34.
In the Introduction, would be great if you could provide more studies on Vitamin D and COVID-19 worldwide.
We added more references with the background of our study in the introduction as you requested and this can be seen in line 45-46; 62-68.
In section 2, you have to justify your sample size regarding representativeness. 148 patients from only one clinic, in my point of view, is an important limitation of your research, especially considering you are submitting this study to a Q1 international journal.
It is a limitation and we added extra sentence related to this as well see in line 370-371. However, our sample size is comparable to those roughly relevant original article and even RCT published in this topic.Ref: Domazet Bugarin, Josipa et al. “Vitamin D Supplementation and Clinical Outcomes in Severe COVID-19 Patients-Randomized Controlled Trial.” Nutrients vol. 15,5 1234. 28 Feb. 2023, doi:10.3390/nu15051234
The Discussion can also be improved and more data from other studies conducted in other regions of the world should be mentioned.
We modified in a significant extent the discussion section with more aspects and additional reference and their details but shorten the discussion section as requested by the other reviewers and cut back unnecessary repetitions as well. See in line 301-303; 318-320; 342-368.
A Conclusions section is missing. Future perspectives should also be provided.”
We added more specific conclusion with future perspective at the end of the Summary section as can be seen in line 387-389.
Thanks to the reviewer for his work in highlighting the weaknesses of the original manuscript. After correcting these, we hope to consider our work worthy of publication.

Round 2
Reviewer 2 Report
Comments and Suggestions for Authors
Thank you to the authors for their responses, which have allowed the originality of this manuscript to be reflected and discussed.
Author Response
Dear Reviewer,
Thank you for your thoughts, and thank you for accepting our manuscript for publication.
Kind regards
Reviewer 3 Report
Comments and Suggestions for Authors
I'm satisfied with the improvements and answers provided by the authors.
Author Response

(The authors gave the same response as above.)
